# NN-LCS: Neural Network and Linear Coordinate Solver Fusion Method for UWB Localization in Car Keyless Entry System

**DOI:** 10.3390/s23052694

**Published:** 2023-03-01

**Authors:** Zengwei Zheng, Shuang Yan, Lin Sun, Hengxin Shu, Xiaowei Zhou

**Affiliations:** 1College of Computer and Computing Science, Hangzhou City University, Hangzhou 310015, China; 2College of Computer Science and Technology, Zhejiang University, Hangzhou 310007, China

**Keywords:** car keyless entry system, none-line-of-sight, UWB localization, ranging error correction

## Abstract

Nowadays, ultra-wideband (UWB) technology is becoming a new approach to localize keyfobs in the car keyless entry system (KES), because it provides precise localization and secure communication. However, for vehicles the distance ranging suffers from great errors because of none-line-of-sight (NLOS) which is raised by the car. Regarding the NLOS problem, efforts have been made to mitigate the point-to-point ranging error or to estimate the tag coordinate by neural networks. However, it still suffers from some problems such as low accuracy, overfitting, or a large number of parameters. In order to address these problems, we propose a fusion method of a neural network and linear coordinate solver (NN-LCS). We use two FC layers to extract the distance feature and received signal strength (RSS) feature, respectively, and a multi-layer perceptron (MLP) to estimate the distances with the fusion of these two features. We prove that the least square method which supports error loss backpropagation in the neural network is feasible for distance correcting learning. Therefore, our model is end-to-end and directly outputs the localization results. The results show that the proposed method is high-accuracy and with small model size which could be easily deployed on embedded devices with low computing ability.

## 1. Introduction

Car keyless entry system (KES) has been emerging as a new trend of car accessing solution [1,2]. Drivers can conveniently access and start the car with a digital key on their smartphone. One important issue of the KES is how to accurately locate the driver’s smartphone to provide a welcome message when approaching the car, unlocking the door when nearing the car, and locking when leaving the car [3]. The proper operation of those functions needs to obtain the accurate position of the keyfob. For example, when the user is standing outside the driver’s door attempting to open it, the KES needs to accurately give a localization result to prove that the keyfob is outside the vehicle, which requires the localization error to be lower than 30 cm. Currently, most existing KESs employ Bluetooth low energy (BLE) to localize keyfobs. BLE’s distance ranging is based on the signal strength and its limitation is that the estimated distance between the BLE nodes is quite sensitive to the environment. Nowadays, ultra-wideband (UWB) technology is becoming a new approach of localizing keyfobs in the KES because it provides precise localization and secure communication [4,5].

However, for vehicles the distance ranging shows significant errors because of none-line-of-sight (NLOS) raised by the car. Figure 1 shows the NLOS challenge in the KES. The five-anchor configuration is popular in the KES [6], where four anchors are at the corners of the car and the fifth is inside. The KES always uses the time-of-flight (ToF) method [7] to measure the distance between the keyfob and the anchors. The anchors, which are at the sides of the keyfob, i.e., $d_{2}$ and $d_{3}$, are on the direct line-of-sight (LOS) path, yielding accurate ToF distances. Therefore, $d_{1}$, $d_{4}$, and $d_{5}$, are on the NLOS path. The table in Figure 1 shows real distances, ToF distances, and relative errors. We find that the errors of the NLOS paths are more than 40%. Therefore, the distances of the NLOS paths can lead to large positioning errors in localizing the keyfob.

In this paper, we propose a novel neural network and linear coordinate solver fusion method (NN-LCS) to mitigate errors caused by the NLOS paths, resulting in localizing keyfobs accurately. The proposed method is an end-to-end learning architecture. The model extracts the localization information from the ToF distances and received signal strength (RSS) measurements to estimate the corrected distances and then calculate the coordinates of the keyfob as the output. We obtain state-of-the-art performances in the localization task in the KES. The contributions of this paper are as follows:This study demonstrates the possibility and effectiveness to embed a localization algorithm in a neural network. The results show that it is better to train the model by minimizing the weighted sum of ranging error and localization error than by minimizing either the ranging error only or the localization error only.The parameter number of the proposed method is very small. Hence, it is edge-affordable both spatially and temporally and can be deployed in cheap microcontrollers (MCU) with low computing ability and small memory.

## 2. Related Work

UWB technology provides an excellent means for wireless positioning due to its high-resolution capability in the time domain. The fundamental mechanisms for localization can apply to all radio–air interfaces, which includes angle-of-arrival-(AOA-), time-of-flight-(ToF-), time-difference-of-arrival-(TDOA-) [8], and RSS-based methods. However, due to the high time resolution of UWB signals, time-based ranging schemes usually provide better accuracy than other mechanisms [9]. However, time-based UWB ranging and localization methods still suffer from NLOS errors, which is especially serious in the KES scenario. In the past, efforts have been made to mitigate the ranging and localization error caused by the NLOS. Generally, we divide previous studies into numerical methods, deep learning methods estimating ranging error, and deep learning methods estimating coordinates.

Many numerical algorithms have been proposed to calculate the tag positions [10,11,12]. For example, Tomic et al. [10] proposed an algorithm for localization in an NLOS environment utilizing RSS and range derived by converting the original non-convex problem into a generalized trust region sub-problem framework, which can be solved exactly by a bisection procedure. However, using numerical algorithms to solve this problem always results in approximation errors. In addition, the parameters in RSS and TOA measurement models are affected differently in different environments.

Some researchers applied deep learning models to ranging error estimation with the help of channel indicators such as RSS, received azimuth and elevation. Krapevz et al. [13] proposed a neural network to estimate the distance by ToF-measured distances, azimuth and elevation. Shalihan et al. [14] use a neural network to estimate the probability of ranging measurements being in the LOS, which were then used as the weights in the weighted least square localization method. Deep learning models were also applied to ranging error estimation exploiting channel impulse response (CIR) data as the input for deep networks to acquire the NLOS condition in the environment [15,16]. In these works, deep neural networks are designed to mitigate the NLOS ranging error. CNN-DE [17] was proposed using a sequence of CIR measures selected as the input to estimate the corrected distances. Simone et al. [15] proposed an efficient representation learning methodology, exploiting the latest advancements in deep learning and graph optimization techniques to achieve effective ranging error mitigation at the edge. CIR signals were directly exploited to extract high semantic features to estimate corrections in either NLOS or LOS conditions. Chengzhi et al. [18] introduced a probabilistic learning approach to mitigate the ranging error and yield uncertainties using the CIR feature. Kim et al. [16] considered the NLOS problem as a 10-class classification problem. For each anchor, a classification result corresponded to an average error and an error variance according to the statistics. These characteristics were used in an extended Kalman filter (EKF) to predict coordinates. The above methods chose the ranging error as the optimizing target. The drawbacks include isolating the ranging errors and ignoring the correlation among the distances from different anchors. In addition, the CIR has some disadvantages: (1) CIR is not available for some low-cost (<$100 each) UWB modules such as the LinkTrack-S UWB module used in [19]. (2) The distribution of CIR changes with the environment [20], which means the features are unstable and not robust; therefore, CIR is not suitable for the KES scenario in which the vehicle transfers from place to place with uncertainty. (2) Compared to other channel-related indicators, eg. ToF and RSS, CIR is more sensitive to environmental changes because it contains all the information of the detailed multi-path in the environment. Different reflective surface compositions or any surface moving in the region result in changes in CIR components, which happens very frequently in a vehicle, e.g., different parking lot or pedestrian walking by. (3) CIR is high-dimensional (typically 100-dimensions), so the CIR feature pattern is much more complex than ToF, and needs a large-scale neural network for feature extraction, making it impossible to be used in the cheap MCUs.

To achieve end-to-end training in the localization task, some studies proposed methods based on deep neural networks (DNN) [19,21,22,23], which directly optimized the tag coordinate error as the network output and final localization coordinate. For example, Jie et al. [23], Li et al. [24] and Poulose et al. [25] proposed different neural networks to estimate coordinates by ToF distances from every anchor to the tag. This type of work does not work well in the vehicle environment because the NLOS is much worse. To increase precision, Yang et al. [19] proposed an end-to-end deep neural network with both distance and RSS measurements. The CNN module, LSTM module, and fully-connected layers were presented to extract the local spatial and temporal features between consecutive frames and estimate the 3D positions, respectively. Nguyen et al. [21] and Nosrati et al. [22] used raw and manually extracted CIR features. CNN-LE [17] was proposed using a sequence of CIR measures selected as the input to estimate the corrected distances. These studies tried to train the model to fit the localization task. However, the model’s huge number of parameters resulted in low feasibility in practice, especially in vehicle scenarios that required algorithms to run locally (offline) with low latency on low-power MCUs. For example, Fontaine et al. [26] used an NVIDIA Jetson Nano GPU for edge inference. Angarano et al. [15] tested their models in GPUs, edge TPUs and high-performance edge CPUs, showing the minimal consumption using float16 needs 32.7 KB flash, and costs a Cortex-A53 (four-core, floating point unit (FPU) inside, 800 MHz) 11.2 ms for network forward inference. However, none of the multi-core, FPU or high frequency are accessible in typical automotive-grade MCUs, e.g., Renesas RH850(@80 MHz max) used in our system, so time consumption could be hundreds of times more.

Related studies are summarized in Table 1. Compared to the methods estimating the ranging errors or NLOS conditions, the methods estimating localization error are the best end-to-end approach in the localization task [19]. The disadvantage of the neural network methods estimating the localization error is the large number of parameters of neural networks and are not suitable for use in embedded MCUs. Thus, we have to develop a method that can reduce the size of parameters while maintaining a good localization performance. In this work, we (1) use D+R pairs as the features which contain both distance and signal strength information to yield features with a small dimension; (2) embed LSM in the neural network model, because we think the linear layers used to estimate coordinates in Yang et al. [19] and Poulose et al. [25] result in a large number of parameters.

## 3. Method

Concerning the existing deep learning methods, fitting the ranging error and the localization coordinates both need a massive number of parameters; thus, we propose a method that could make use of a linear coordinate solver for the localization to address the existing problems. In this section, we overview the UWB localization task in the KES (in Section 3.1) for the problem statement and symbols definition, then we present the NN-LCS model (in Section 3.3), introducing the model’s architecture which is combined with a distance correction network (DCNN) module for extracting the ToF measurement and RSS measurements features to calculate a closer estimation of the actual distances (corrected distances), and a LSM localization algorithm module to calculate the localization coordinates from the corrected distances. The calculation process and backpropagation feasibility of the LSM algorithm will also be covered.

### 3.1. Task Overview

The task of the keyfob localization in the KES needs accurate positioning and automatically recognizes the mobile when the driver is approaching, unlocks the door within two meters, or locks the door when the driver is leaving. Concerning the KES requirements, a Cartesian coordinate system was selected with its *x* axis to the right of the car, its *y* axis to the front of the car, and its *z* axis vertically up. The centre point of the circumscribed rectangle of the car’s vertical projection to the ground was selected as the origin.

The coordinates of anchors A are previously measured or calibrated manually.
(1)$$A={\left[a_{1},a_{2},\ldots ,a_{N}\right]}^{T}=\left[\begin{array}{ccc} a_{1}^{x} & a_{1}^{y} & a_{1}^{z}\\ a_{2}^{x} & a_{2}^{y} & a_{2}^{z}\\ \vdots & \vdots & \vdots \\ a_{N}^{x} & a_{N}^{y} & a_{N}^{z} \end{array}\right],$$
where *N* is the total amount of UWB anchors, and $a_{i}$ is the coordinate of anchor *i*.

In distance ranging and correction, we denote the ground true distance as $d={\left[d_{1},\ldots ,d_{i},\ldots ,d_{N}\right]}^{T}$, the measured ToF distance as $d^{t}={\left[d_{1}^{t},\ldots ,d_{i}^{t},\ldots ,d_{N}^{t}\right]}^{T}$, and the corrected ToF distance as ${\hat{d}}^{t}={\left[{\hat{d}}_{1}^{t},\ldots ,{\hat{d}}_{i}^{t},\ldots ,{\hat{d}}_{N}^{t}\right]}^{T}$ where i∈{1,2,…,N} is the index of an anchor and *N* is the total amount of UWB anchors.

In localization computation, we denote the estimated coordinates of the keyfob as $\hat{p}={\left[\hat{x},\hat{y},\hat{z}\right]}^{T}$ and the ground true coordinates as $p={\left[x,y,z\right]}^{T}$. Therefore the localization error (LE) Δp is defined as,
(2)$$\Delta p=LE\left(p,\hat{p}\right)=∥p-\hat{p}∥,$$
where $∥\cdot ∥$ is the Euclidean distance.

For a localization dataset, the mean localization error (MLE) metric, which is defined as the mean value of LE Δp over *M* testing points, is used to evaluate the overall performance of the localization. The central goal of the localization task is to minimize the MLE metric.
(3)$$MLE=\frac{1}{M}\cdot \sum_{i=1}^{M}\Delta p\left[i\right],$$
where $\Delta p\left[i\right]$ is the LE of testing point *i*.

### 3.2. ToF Ranging Method

The distances between keyfob–anchor pairs are obtained using alternative double-sided two-way ranging (ADS-TWR) as described in [27]. The main advantage of ADS-TWR is the measurement of the time of flight (ToF) between two UWB devices without the need for additional clock synchronization. However, compared to other ranging methods, such as TOA or TDOA-based approaches, more communication between the UWB devices is required, leading to a higher bandwidth utilization which decreases the measurement rate [28].

Figure 2 summarizes the communication workflow for a single keyfob–anchor pair. The ADS-TWR is initiated by a poll message sent from the keyfob addressing an anchor with a specific anchor ID. After receiving the replied response message, the keyfob transmits the final message. The round time between transmitting the poll and receiving the response $t_{rd1}$ is logged in the keyfob. On the anchor side, the reply time between receiving the poll and transmitting the response $t_{rp1}$ is logged on the anchor. The round time between transmitting the response and receiving the final $t_{rd2}$, and the reply time between receiving the response and transmitting the final $t_{rp2}$ are logged in the anchor and keyfob, respectively. In the final message, the keyfob transmits $t_{rd1}$ and $t_{rp2}$ to the anchor. On anchor side, supposing the anchor’s ID is *i*, the MCU computes the time of flight $t_{i}^{t}$ for the keyfob and ToF distance by:(4)$$\begin{array}{c} t_{i}^{t}=\frac{\left(t_{rd1}\times t_{rd2}\right)-\left(t_{rp1}\times t_{rp2}\right)}{\left(t_{rd1}+t_{rd2}+t_{rp1}+t_{rp2}\right)},\\ d_{i}^{t}=c\cdot t_{i}^{t}, \end{array}$$
where *c* is the speed of light.

Then a message containing anchor ID *i*, ToF distance $d_{i}^{t}$ and RSS of the final message $r_{i}^{t}$ is transmitted on the CAN bus. A central MCU node, termed the host, will parse the frames for the localization task.

### 3.3. NN-LCS Model

As shown in Figure 3, the NN-LCS model contains a distance correction neural network (DCNN) and an LSM algorithm module. DCNN extracts the features and estimates the actual distances. The LSM algorithm module calculates the coordinates of the keyfob from the estimated distances.

#### 3.3.1. DCNN

Suppose the number of anchors is *N*, two *N*-dimensional vectors, ToF ranging results $d^{t}$ and RSS measurements $r^{t}$ are selected as the input of the DCNN model. A fully-connected (FC) layer $FC_{1}$ is used to extract the distance feature $f_{1}$ from the ToF distances $d^{t}$. Note that in all FCs below, LeakyRELU (slope = 0.1) [30] is chosen as the activation function,
(5)$$\begin{array}{c} f_{1}=LeakyRELU\left(FC_{1}\left(d^{t}\right)\right). \end{array}$$

$FC_{2}$ is used to extract the RSS features $f_{2}$ from the RSS measurements $r^{t}$. The Softmax function is performed on $r^{t}$ to convert the RSS (db) to power and normalize to [0, 1] which is denoted by $r_{\mathrm{sm}}^{t}$ and used as the input of $FC_{2}$. We add the RSS features to the neural network because the RSS value can also imply the topological relationship between the keyfob and the anchors,
(6)$$\begin{array}{c} r_{\mathrm{sm}}^{t}=Softmax\left(r^{t}\right),\\ f_{2}=LeakyRELU\left(FC_{2}\left(r_{\mathrm{sm}}^{t}\right)\right). \end{array}$$

Then a feature concatenation is performed between $f_{1}$ and $f_{2}$ as the input of the next neural layer,
(7)$$f=f_{1}⊕f_{2}.$$

The feature f is then put into a multi-layer perceptron MLP for distance estimation to obtain the corrected ToF distances ${\hat{d}}^{t}$, and ${\hat{d}}^{t}$ is the input into the LSM localization algorithm to calculate the localization results $\hat{p}$.
(8)$${\hat{d}}^{t}=MLP\left(f\right)$$
(9)$$\hat{p}=LSM\left({\hat{d}}^{t}\right).$$

Finally, the errors of the localization coordinate $\hat{p}$ and corrected distances ${\hat{d}}^{t}$ are calculated and backpropagated to optimize the parameters of the DCNN.

#### 3.3.2. LSM Localization Algorithm

LSM is a well-known algorithm in localization tasks. The detailed 2D and 3D LSM derivation can be found in [31] and [23]. In LSM derivation, a matrix equation is constructed *C* for solving p,
(10)$$\begin{array}{c} C\cdot \hat{p}=b\left({\hat{d}}^{t}\right)\\ C=-2\cdot {\left[\begin{array}{c} a_{2}-a_{1},a_{3}-a_{1},\ldots ,a_{N}-a_{1} \end{array}\right]}^{T} \end{array}$$
(11)$$\begin{array}{c} b\left({\hat{d}}^{t}\right)=m\left({\hat{d}}^{t}\right)\circ m\left({\hat{d}}^{t}\right)-n\left({\hat{d}}^{t}\right)\circ n\left({\hat{d}}^{t}\right)-s\\ m\left({\hat{d}}^{t}\right)=\left[\begin{array}{cc} 0_{(N-1)\times 1} & I_{\left(N-1)\times (N-1\right)} \end{array}\right]\cdot {\hat{d}}^{t}\\ n\left({\hat{d}}^{t}\right)=\left[\begin{array}{cc} 1_{(N-1)\times 1} & 0_{\left(N-1)\times (N-1\right)} \end{array}\right]\cdot {\hat{d}}^{t}\\ s={\left[\begin{array}{c} {∥a_{2}∥}^{2}-{∥a_{1}∥}^{2},{∥a_{3}∥}^{2}-{∥a_{1}∥}^{2},\ldots ,{∥a_{N}∥}^{2}-{∥a_{1}∥}^{2} \end{array}\right]}^{T}, \end{array}$$
where the estimated coordinate $\hat{p}$ is the solution and ∘ indicates the Hadamard product (element-wise multiplication). Thus, the coordinates of the keyfob can be calculated as
(12)$$\begin{array}{c} \hat{p}=LSM\left({\hat{d}}^{t}\right)=P\cdot b\left({\hat{d}}^{t}\right)\\ P={\left(C^{T}\cdot C\right)}^{-1}\cdot C^{T}, \end{array}$$
the estimated coordinate $\hat{p}$ is the solution.

Because the LSM, a linear solver for quadratic optimization problems with well-known convergence properties, is embedded in the NN-LCS, we have to confirm that it is possible to train a neural network combined with the LSM algorithm. Duong et al. [32] proved that the cascade of a NN (that converges by hypothesis) and an LSM converge as well. Therefore, we only need to make sure that the conversion from ${\hat{d}}^{t}$ to b, as shown in Equation (Equation 11), also supports error backpropagation, which is easy as the derivative is calculated in Equation (Equation 13).
(13)$$\begin{array}{cc} \frac{\partial b_{i}}{\partial {\hat{d}}_{j}^{t}} & =\left\{\begin{array}{cc} =-2{\hat{d}}_{i}^{t}, & j=1\\ =2{\hat{d}}_{i}^{t}, & j\in \{2,3,\ldots ,N\},i=j-1.\\ =0, & j\in \{2,3,\ldots ,N\},i\neq j-1 \end{array}\right. \end{array}$$
where i∈{1,2,…,N−1}, j∈{1,2,…,N},

#### 3.3.3. Loss Function

If we only use the localization error $L_{1}\left(\hat{p}\right)$, the model will overfit the training set and generate unreasonable corrected distances, such as higher than 4 m. Therefore, we design a combined loss function with localization error and distance error to train the model and use the TDMAE as described in Equation (Equation 17) in the loss function. Distance weight α is introduced to balance the influence of localization error and distance error,
(14)$$L_{1}\left(\hat{p}\right)=LE\left(p,\hat{p}\right)$$
(15)$$L_{2}\left({\hat{d}}^{T}\right)=TDAE\left(d,{\hat{d}}^{T}\right)$$
(16)$$L\left(\hat{p},{\hat{d}}^{T}\right)=\left(1-\alpha \right)\cdot L_{1}\left(\hat{p}\right)+\alpha \cdot L_{2}\left({\hat{d}}^{T}\right).$$

### 3.4. Metrics

As described in Section 3.1, MLE is used as a metric to evaluate the overall localization error. Furthermore, to evaluate the overall ranging error, ToF distance absolute error (TDAE) is defined in Equation (Equation 17). TDMAE is defined as the mean value of TDAE over the testing points.
(17)$$\begin{array}{c} \Delta d=TDAE\left(d,d^{t}\right)=\frac{1}{N}\cdot \sum_{i=0}^{N}\left|d_{i}^{t}-d_{i}\right|,\\ TDMAE=\frac{1}{M}\cdot \sum_{i=1}^{M}\Delta d\left[i\right], \end{array}$$
where $\Delta d\left[i\right]$ is the TDAE of testing record *i*, *M* denotes the number of testing records, *N* denotes the number of anchors, and *d* and $d^{t}$ denote the actual and estimated distances, respectively.

## 4. Experimental Settings

### 4.1. System Overview

For evaluation, we built an on-vehicle localization system, in which the anchors were configured as shown in Figure 1 and the workflow is shown in Figure 4. In the system, the UWB ranging was based on Decawave DW1000 chipset with a data rate of 110 Kbps, channel 2, indicating a bandwidth of 3774.0–4243.2 Mhz, and a preamble code frequency of 64 MHz and length 2048. The collecting system was built on Renesas RH850, which is a single-core chip with no float point unit (FPU) and a main frequency of 80 MHz. No extra memory or storage chip was used except the integrated internal 512 KB flash and 32 KB RAM. The anchors and keyfob hardware are shown in Figure 5. The Anchors were connected by a wire for power and CAN communication, on which the ranging results and RSS were uploaded to the host MCU and then transmitted to a PC.

### 4.2. Dataset

We drew rounded rectangles on the ground around the testing car for positioning evaluation, where contour 1 was 2 m from the bounding rectangle of the car, contour 2 was 4 m, and contour 3 was 6 m, as shown in Figure 6. 12 points on each contour were used as the collecting points. On each point, the data were collected within 6 s, i.e., 60 records. It took three students one day to collect the dataset on a prototype vehicle. Finally, we collected 2160 records in total. Note that we performed a moving average filter (window = 10) on the ToF ranging measurements to filter noise. Ground true localization coordinates were manually measured by a tape.

For dataset splitting, we applied *k*-fold evaluation (k=6). We split the dataset according to the collecting points, helping prevent overfitting. Out of *N* collecting points $p_{1},p_{2},\ldots ,p_{N},\left(N=36\right)$, we randomly divided them into *k* different subsets, each of which consists of data from N/k=6 out of the *N* points, as shown in Table 2. For each fold, one subset is used for testing and the others for training. All the *k* testing predict errors are stored and merged as the *k*-fold result of a method. The results of all methods shown in Section 5 were obtained in this way.

### 4.3. Model Training

We used a 3D coordinate as the model’s output only in the training process. However, according to the actual requirements, the height of the keyfob did not need to be considered, since the KES works once the user approaches the car no matter if the keyfob is held over head or lying in a bag. Therefore, we only considered the first two dimensions ${\hat{p}}^{\prime }={\left[\hat{x},\hat{y}\right]}^{T}$ in the testing process, which means all the localization errors in Section 5 are 2D errors. An Adam optimizer [33] was utilized for training owing to its high effectiveness in handling stochastic objective functions [19].

Hyperparameters and settings for the NN-LCS and comparative methods are listed in Table 3. We carried out parameter adjustment experiments reported by Yang et al. [19] for most of the hyperparameters including the network layer and neuron numbers. The adjustment did not improve the performance significantly, so we used the original settings from the article. From Tomic et al. [10], we carried parameter fitting experiments for γ values and selected a LOS sequence to acquire the $d_{0}$ and $P_{0}$ pair.

## 5. Results

### 5.1. Performance of Localization Error

We compare our NN-LCS with the numerical methods of Tomic et al. [10], and the spatial–temporal deep neural network proposed by Yang et al. [19]. Among the two comparative studies, the method proposed by Tomic et al. is a state-of-the-art numerical approach involving both RSS and distance measurements. In addition, Yang et al. is a state-of-the-art deep learning-based scheme with both RSS and distance measurements.

We use a boxplot to show the performance of the localization at the three contours, shown in Figure 7. We find that: (1) The Yang et al. method shows the biggest localization error on our dataset. This could be because of too much overfitting on the points in the training set. (2) Both of the two competing methods show significant rising trends as the distance between the keyfob and the vehicle increases. As shown in Figure 8, the original absolute NLOS error (TDMAE) at the three contours is similar. This means that the comparative methods become more sensitive to the NLOS error at further testing points. (3) The proposed NN-LCS method shows minimum localization error on all three contours. Furthermore, the localization error does not change significantly at the different contours.

### 5.2. Performance of the Ranging Error Correction

The cumulative distribution function (CDF) in Figure 8 shows the comparison between the original ranging results and the corrected results by our model in TDAE. In the original raw distance data, as the chain lines show, more than 40% of the error on each contour could exceed 0.5 m and more than 20% of the error on contour 1 could exceed 1 m which is a big threat to accurate decimetre localization. However, as the solid lines show in Figure 8, after the process of the DCNN in our model, the corrected distances show much less error in TDAE with at least 80% of the error on each contour less than 0.25 m. This proves the effectiveness of the DCNN and explains why the NN-LCS could work properly.

Compared to the existing methods that estimate the ranging errors, the NN-LCS also shows the ability to correct the ToF distances. The TDMAE of the original ToF distance in our dataset is 0.7437 m and the corrected distances by the DCNN obtain much less error TDMAE = 0.1804 m, i.e., 75% lower than that of the original distances. We also evaluated our method in the RMSE (root-mean-square-error) metric to compare it with the performance of the research published by Chengzhi et al. [18], which used CIR signals and collected the data in an indoor environment. The RMSE results of our NN-LCS and Chengzhi et al. [18] are 0.18 m and 0.26 m, respectively. Moreover, the NLOS condition in the car is much worse outdoor than indoors.

### 5.3. Ablation Results

In order to investigate the effects of distance and RSS in the proposed method, we designed two network architectures with only distance for the input (or RSS only) as the ablation experimental groups. The results of the ablation experiment are shown in Table 4.

In general, the RSS information does not contain the information needed for localization, as the ranging error could reach more than 1 m. For contours 2 and 3, as well as the whole dataset, the model with distance and RSS performs the best with the fusion of the RSS information, which proves that the feature extraction reduces the localization error. However, for contour 1, the model with distance only performs better than that with distance and RSS. Regarding the user manual of the DW1000 chipset, the RSS level estimation shows obvious non-linearity and instability because of saturation, so the overtake of the model with distance only on contour 1 is very likely to be because of measurement saturation of DW1000’s receiver.

### 5.4. Analysis of Distance Loss Weight

We carry out the experiment on distance loss weight α described in the loss function Equation (Equation 16). As shown in Figure 9, with the increase in α, the TDMAE decreases rapidly, and overall the MLE also decreases.

An extreme circumstance when α is set to be 1.0 and the model is actually set to fit the ranging error only, results in an MLE error bigger than the setting with α set to 0.9, which disapproves the assumptions made by studies minimizing distance errors [15,26]. When α is set to be 0 and the model is actually set to fit the localization error only, the model does not show the best performance on both MLE and TDMAE. Moreover, this setting leads the biggest TDMAE error of more than 4 m, which is even bigger than the TDMAE of the original data, thus bringing unreasonable localization results.

The best MLE performance of the model is when α is set to be 0.9, with MLE =0.29 m and TDMAE =0.12 m. This infers that the correlation between the ranging error and the localization error is not as simple as it seems to be, and suggests that neither minimizing the ranging error nor minimizing the localization error is the best way for localization tasks. The best training methodology is to minimize a combination of these two errors.

In addition, this experiment proves that the proposed model is robust in the α hyperparameter setting, because no matter what α is set to, the model could perform better than the comparative methods on the MLE metric.

## 6. Conclusions and Future Works

In order to solve the localization task in the car KES, this paper proposed a novel end-to-end learning architecture. The NN-LCS model uses a DCNN which extracts the localization information from the ToF distances and RSS measurements to estimate the corrected distances and then, by a linear coordinate solver, calculates the coordinates of the keyfob as the output. This study demonstrates the possibility to embed a localization algorithm in a neural network. The results show that it is better to train the model by minimizing the weighted sum of the ranging error and localization error than by minimizing either the ranging error only or the localization error only. Furthermore, the number of parameters of the proposed method is extremely small. Hence, it is edge-affordable both spatially and temporally and can be deployed in cheap MCUs with low computing ability.

According to the experiments in Section 5, state-of-the-art NN-LCS localization accuracy is achieved in the KES scenario. Its accuracy in distance correction proves the effectiveness of the DCNN module in the model and explains why NN-LCS could work well. Moreover, the ablation results show the validity of fusion between distances and RSS. Finally, the analysis of distance loss weight reveals the fact that compared to training the NLOS error mitigation model by minimizing the distance error only or localization error only, it is better to train by minimizing the combined loss function with the localization error and distance error.

However, our current work collected isolated point data for model training and testing, which is low in collecting efficiency and needs to be verified in other scenarios. In the future, we will collect trajectory data based on LiDAR-SLAM [34] or VSLAM [35,36] to enable low-cost and large-scale sample collection. In addition, we will introduce more improved algorithms instead of LSM into the NN-LCS to achieve better accuracy and robustness. Further more, although the CIR signals show low stability and high dimensionality at present, we will try to build specific neural networks that can remove the uncertainty and extract the CIR features with minimal cost.

## Figures and Tables

**Figure 1 sensors-23-02694-f001:**
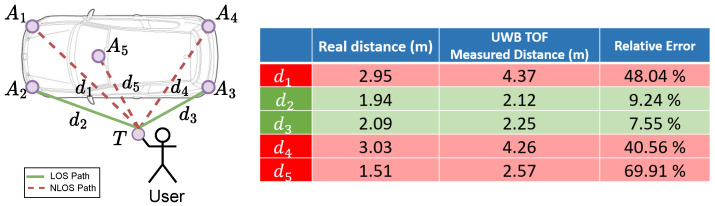
Illustration of the NLOS challenge in the KES. The solid lines $d_{2}$ and $d_{3}$ denote the LOS paths and the dashed lines $d_{1}$, $d_{4}$, and $d_{5}$ denote the NLOS paths. The estimated position of the keyfob can be greatly affected by distances $d_{1}$, $d_{4}$, and $d_{5}$, i.e., more than 40% relative error.

**Figure 2 sensors-23-02694-f002:**
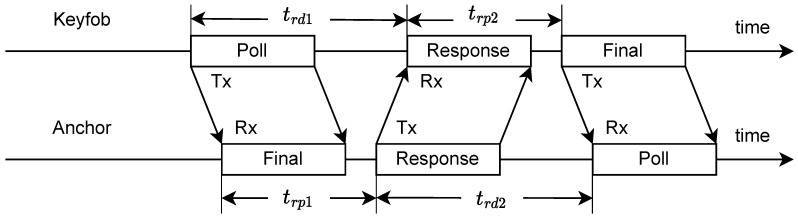
Illustration of the ADS-TWR ranging workflow [29].

**Figure 3 sensors-23-02694-f003:**
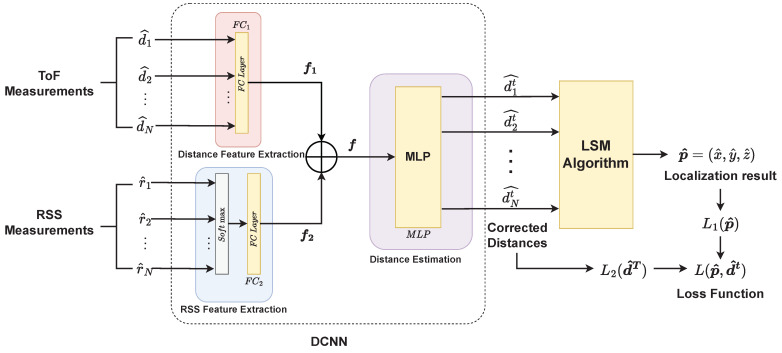
The architecture of the NN-LCS model.

**Figure 4 sensors-23-02694-f004:**
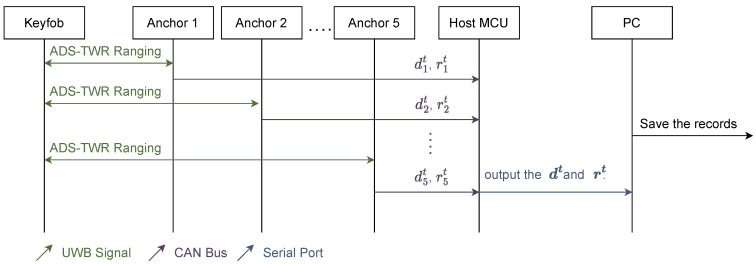
The flow chart of the localization procedure in our KES system.

**Figure 5 sensors-23-02694-f005:**
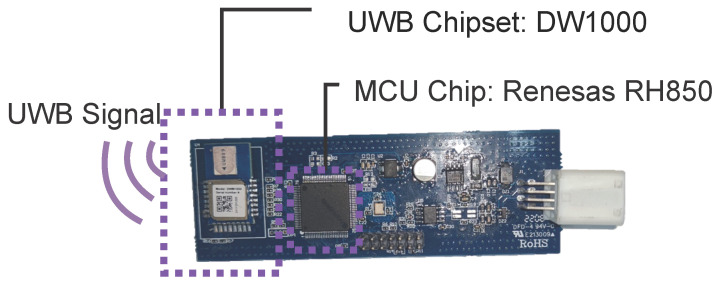
Illustration of the UWB anchors and the keyfob hardware.

**Figure 6 sensors-23-02694-f006:**
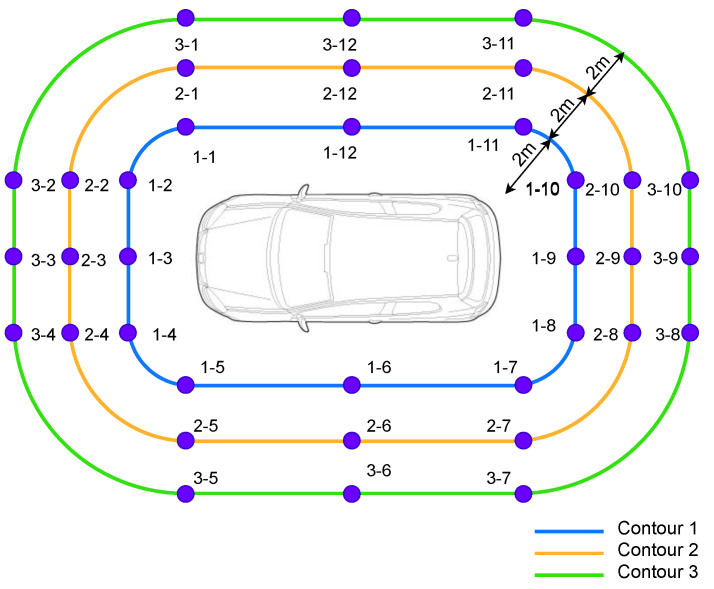
Data collecting points. The collecting points are labelled as purple dots. Contour 1 was 2 m from the car’s bounding rectangle, contour 2 was 4 m from the car’s bounding rectangle, and contour 3 was 6 m from the car’s bounding rectangle.

**Figure 7 sensors-23-02694-f007:**
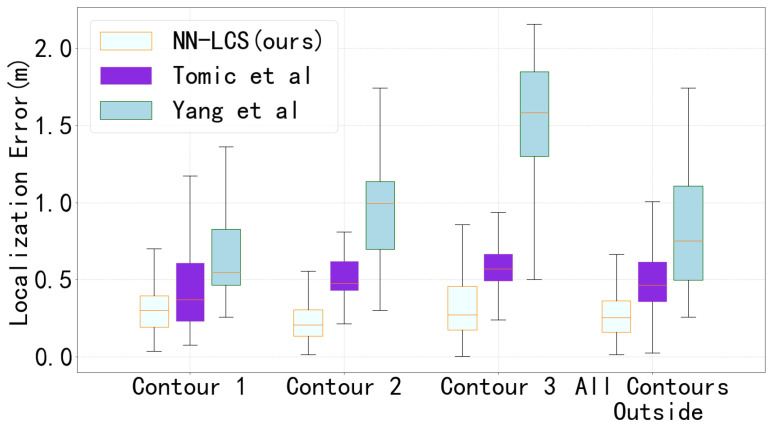
Boxplot of the localization error comparison (m). The test points are contour 1 (1-1,1-2…,1-12), contour 2 (2-1,2-2…,2-12), and contour 3 (3-1,3-2…,3-12).

**Figure 8 sensors-23-02694-f008:**
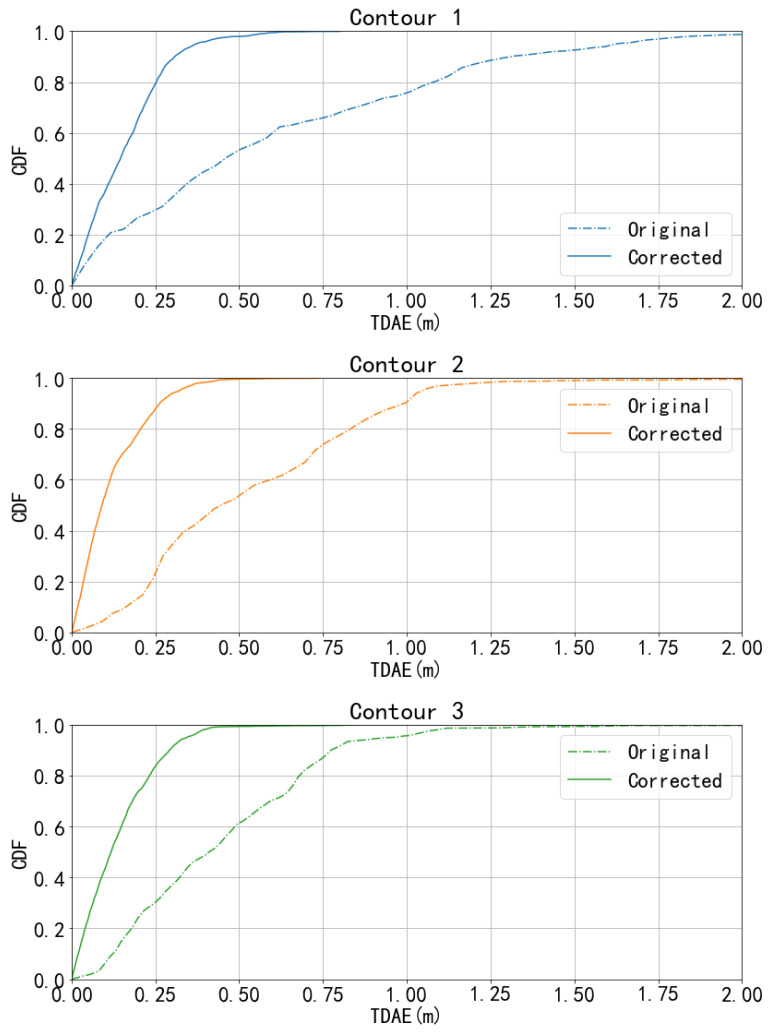
Comparison of the cumulative distribution function (CDF) between the original ranging results and the corrected results by our model in TDAE. The chain and solid lines represent the original ranging errors and corrected distance errors, respectively. Original refers to the error of the raw ToF distances in the dataset, denoted as d in Figure 3, and Corrected refers to the errors of the respective corrected distances calculated by DCNN, denoted as $d^{t}$ in Figure 3.

**Figure 9 sensors-23-02694-f009:**
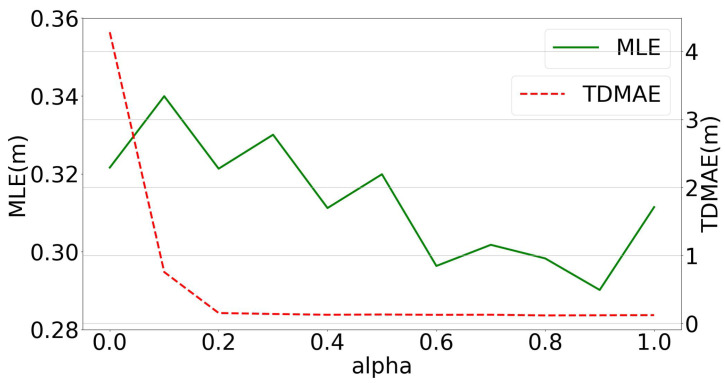
The MLE and TDMAE with the change in distance weight α.

**Table 1 sensors-23-02694-t001:** Comparison of the related methods.

Method	Features	Model	Localization Algorithm	Parameters #	Optimizing Target
Tomic et al. [10]	D ^1^ + R ^2^	-	Numerical	-	-
Tiwari et al. [12]	D + R	-	Numerical	-	-
Kim et al. [16]	CIR	LSTM	LSM and EKF	1 M	NLOS condition Classification
Li et al. [24]	D	NN	NN/LSM	3 K	NLOS condition Classification
Shalihan et al. [14]	D + R	NN	WLS ^4^	30–900 K	Probability of LOS
Krapevz et al. [13]	D, azimuth and elevation	NN	-	31	Ranging Error
Angarano et al. [15]	CIR	CNN	-	6 K	Ranging Error
Fontaine et al. [26]	CIR	CNN	-	32 K	Ranging Error
Chengzhi et al. [18]	CIR	CNN+NPN	-	800 K	Ranging Error
CNN-DE [17]	CIR	CNN	-	24 K	Ranging Error
CNN-LE [17]	CIR	CNN	NN	1 M	Localization Error
Nguyen et al. [21]	CIR	GRU	NN	36 K	Localization Error
Nosrati et al. [22]	CIR-MF ^3^	CNN	NN	300 K	Localization Error
Poulose et al. [25]	D	LSTM	NN	1 M	Localization Error
Yang et al. [19]	D + R	CNN+LSTM	NN	136 K	Localization Error
NN-LCS (ours)	D + R	NN	LSM	2 K	Localization Error

^1^ D denotes ToF distance. ^2^ R denotes RSS. ^3^ CIR-MF denotes manual feature extracted from the CIR. ^4^ WLS is the weighted least square localization method [12].

**Table 2 sensors-23-02694-t002:** Dataset subset split for 6-fold evaluation.

Subset Id	Collecting Points
1	1-3, 1-5, 1-6, 2-2, 2-5, 3-3,
2	1-12, 2-9, 2-11, 3-7, 3-8, 3-9,
3	1-4, 2-3, 3-2, 3-6, 3-11, 3-12,
4	1-1, 1-9, 2-4, 2-6, 2-10, 3-1,
5	1-10, 1-11, 2-7, 2-8, 2-12, 3-5,
6	1-2, 1-7, 1-8, 2-1, 3-4, 3-10,

**Table 3 sensors-23-02694-t003:** Hyperparameters and settings for the NN-LCS and comparative methods. The parameters for the comparative methods listed here are the ones that we modified for the best performance on our dataset.

Parameter Type	Parameter Name	Value
NN-LCS network parameters	$FC_{1}$ size	5×30
	$FC_{2}$ size	5×30
	MLP size	60×30×5
	distance loss weight α	0.9
	LeakyRELU slope	0.1
Training settings (both NN-LCS and Yang et al. [19])	Adam betas	(0.9, 0.999)
	Adam eps	$1\times 10^{-8}$
	Adam weight decay	0
	batch size	4
	max epoch	300
	random seed	37
Training settings (for NN-LCS only)	learning rate	$3\times 10^{-3}$
Training settings (Yang et al. [19] only)	learning rate	$1\times 10^{-3}$
	dropout	0.3
Parameter settings (Tomic et al. [10] only)	γ	0.959
	$d_{0}$	4.25 m
	$P_{0}$	−78.38 dB

**Table 4 sensors-23-02694-t004:** Comparison between models with different features in the MLE(m) metric. Here, D denotes ToF distance and R denotes the RSS.

Input	Contour 1	Contour 2	Contour 3	All
D + R	0.30	0.23	0.34	0.29
D	0.28	0.25	0.39	0.31
R	1.38	0.78	1.97	1.38

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
