# Peer review of "NN-LCS: Neural Network and Linear Coordinate Solver Fusion Method for UWB Localization in Car Keyless Entry System"

_sensors, 2023, doi:10.3390/s23052694_

Round 1
Reviewer 1 Report
The manuscript deals with an active localization method using Ultra Wide-Band (UWB) devices and neural-based pre-processing for car keyless entry system (KES) applications.
The manuscript, except for some language errors, is easily readable. However, I have several remarks mostly regarding the manuscript novelty, its structure, and the mathematical description of the signal model of Sect. 3.
In addition, Sects. 1 and 2 are poor with missing references to important works and state-of-the-art publications.
Sect. 5 is also questionable since the comparison against other methods is incomplete with no description of the parameters adopted for these methods. Without knowing the parameters sets it is not possible to correctly evaluate the results. Some parts of Sect. 5 are obscure and unclear as well.
The bibliography is incomplete as well and relevant references are missing as well.
Specific remarks.
1) There are a huge amount of papers about UWB localization using neural networks. Thus the novelty of the proposed approach is questionable. High-resolution methods are also already available, e.g. [R1] at the end of these comments. For additional references of similar works using UWB and neural network processing-base methods see for instance [R2]-[R7], [10]. Thus the manuscript contributions need to be reconsidered and rewritten.
2) Topics about active localization are more complex than the brief discussion provided in Sect. 2. Please rewrite most parts of this section including also some comments regarding the works discussed in the seminal papers proposed in the Special Issue of the IEEE Signal Processing Magazine, July 2005 and the citing papers hereafter.
3) Lines 66-79: works about UWB localization with neural networks have been available from at least 15 years: e.g. see [R8], [R9]. Please revise this portion of text.
4) Every channel-related indicator (e.g., ToF, RSSI, CIR, CSI) depends on the environment. Please revise and discuss the portion of text at lines 79-83.
5) The problem discussed at line 92 is common to other neural-based methods. Please comment.
6) Lines 96-98 are too generic: please clarify and be more specific.
7) Before eq. (1): where is the origin of the coordinates? Is it relative to the vehicle? Not clear.
8) Line 107-108: check the embedded equations for formal mathematical issues concerning scalar (in italics) and vectors (in bold). Check also superscript and subscript indexes.
9) Line 112: the MLE value is never defined.
10) Lines 116-117: the method called Alternative Double-Sided Two Way Ranging is presented in the paper [R11] where eq. (3) is discussed in more details with respect to the application note [18]. Please use this new reference.
11) Lines 123-131: is the ToF value computed by the nodes? Or by each node? Or by the system? How are the terms of eq. (3) acquired? Not clear, please rewrite.
12) Please check Sect. 3.3.1 for missing variable/vector definition and formal errors.
13) Sect. 3.3.2: the linear Least Squares Multi-lateration algorithm concerning the squared-range differences is well-known and the length of this section can be reduced. In addition, please insert here one or more references related to this algorithm, e.g. the Bensky textbook, and then simplify the equations (some equations have also formal errors as well).
14) Check eqs. (14) and (21) for formal errors.
15) Sect. 3.4: given the results of [R11], since the LSM is a linear system, the cascade of a NN (that converges by hypothesis) and the LSM, converges as well. The proof is trivial given the results of [11]. Please insert the reference, provide a comment and then delete this section.
16) Fig. 4 does not show any system. This figure summarizes the messages exchanged by the system devices. Please revise and add a figure about the proposed localization system.
17) Sect. 4.2 is unclear and confused. Please rewrite.
18) Lines 207-209: this part is unclear. Please revise.
19) For a fair comparison of methods in Sect. 5, the parameters used by methods [8] and [13] must be included. Please add them to Tab. 3.
20) Lines 223-237 and Fig.8: these parts are unclear and must be rewritten.
21) Table 4: D and R are not defined.
22) Sect. 5.1: other localization methods, even high accuracy ones, are already available (see the literature ad the end). For fair comparisons, results must be compared against state-of-the-art methods, e.g. [10], by using optimized parameters for each method.
23) Sect. 5.3 and Tab. 4 are not clear and must be rewritten.
24) The bibliography is poor and many important references are missing. This section must be reconsidered and expanded. See also, but not limited to, the attached list.
25) Reference [5] is not written in English and no English translation is available. Please consider to delete it.
Additional references
[R1] N. C. Rowe, A. E. Fathy, M. J. Kuhn and M. R. Mahfouz, "A UWB transmit-only based scheme for multi-tag support in a millimeter accuracy localization system," 2013 IEEE Topical Conference on Wireless Sensors and Sensor Networks (WiSNet), Austin, TX, USA, 2013.
[R2] P. Krapež, M. Vidmar, and M. Munih, “Distance Measurements in UWB-Radio Localization Systems Corrected with a Feedforward Neural Network Model,” Sensors, vol. 21, no. 7, p. 2294, Mar. 2021.
[R3] Jie, Dong, et al. "A ultra-wideband location algorithm based on neural network." 2010 6th International Conference on Wireless Communications Networking and Mobile Computing (WiCOM). IEEE, 2010.
[R4] Li, B., Zhao, K. & Sandoval, E.B. A UWB-Based Indoor Positioning System Employing Neural Networks. J geovis spat anal 4, 18 (2020).
[R5] M. Shalihan, R. Liu and C. Yuen, "NLOS Ranging Mitigation with Neural Network Model for UWB Localization," 2022 IEEE 18th International Conference on Automation Science and Engineering (CASE), Mexico City, Mexico, pp. 1370-1376, 2022.
[R6] Poulose, A., & Han, D. S. (2020). UWB indoor localization using deep learning LSTM networks. Applied Sciences, Vol. 10, No. 18, 6290.
[R7] Nguyen, D. T. A., Lee, H. G., Jeong, E. R., Lee, H. L., & Joung, J., Deep learning-based localization for UWB systems. Electronics, Vol. 9, No. 10, 1712, 2020.
[R8] A. Taok, N. Kandil, S. Affes and S. Georges, "Fingerprinting Localization Using Ultra-Wideband and Neural Networks," 2007 International Symposium on Signals, Systems and Electronics, Montreal, QC, Canada, pp. 529-532, 2007.
[R9] S. Ergut, R. R. Rao, O. Dural and Z. Sahinoglu, "Localization via TDOA in a UWB Sensor Network using Neural Networks," 2008 IEEE International Conference on Communications, Beijing, China, 2008.
[R10] C. Mao, K. Lin, T. Yu and Y. Shen, "A Probabilistic Learning Approach to UWB Ranging Error Mitigation," 2018 IEEE Global Communications Conference (GLOBECOM), Abu Dhabi, United Arab Emirates, pp. 1-6, 2018.
[R11] D. Neirynck, E. Luk and M. McLaughlin, "An alternative double-sided two-way ranging method," 2016 13th Workshop on Positioning, Navigation and Communications (WPNC), Bremen, Germany, pp. 1-4, 2016.
[R12] Duong, Tuan A., Allen R. Stubberud, "Convergence analysis of cascade error projection-an efficient learning algorithm for hardware implementation." International journal of neural systems 10.03, pp. 199-210, 2000.
Author Response
We appreciate the editor and the reviewers for their valuable comments. The concerns raised by the reviewers have been addressed individually. The changes to the manuscript and our responses to the comments are detailed below. We sincerely hope that the revised manuscript has addressed the raised concerns and is acceptable for publication.
Response to Reviewer 1:
Concern #1: There are a huge amount of papers about UWB localization using neural networks. Thus the novelty of the proposed approach is questionable. High-resolution methods are also already available, e.g. [R1] at the end of these comments. For additional references of similar works using UWB and neural network processing-base methods see for instance [R2]-[R7], [10]. Thus the manuscript contributions need to be reconsidered and rewritten.
Author response:
We appreciate the reviewer’s comments. [R4] and[R5] used neural network to predict an indicator of the NLOS condition for further error mitigation. [R7] proposed two CNN networks using CIR to mitigate the ranging error and the localization error, respectively. [R2] estimate real distances. [R3],[R4], and [R6] estimated tag’s coordinate by NN. None of them have embedded numerical localization algorithm in the network architecture, therefore we think the embedded localization algorithm with neural network is a contribution of our work.
Author action: The recommended references are added in the revised paper and the contributions are revised.
Concern #2: Topics about active localization are more complex than the brief discussion provided in Sect. 2. Please rewrite most parts of this section including also some comments regarding the works discussed in the seminal papers proposed in the Special Issue of the IEEE Signal Processing Magazine, July 2005 and the citing papers hereafter.
Author response: We appreciate the reviewer’s comments. The suggestion is valuable because the article is missing with overview of localization mechanism.
Author action: Those works were added in lines 55-60 for overview of localization mechanism and comment on different mechanisms in UWB systems.
Concern #3: Lines 66-79: works about UWB localization with neural networks have been available from at least 15 years: e.g. see [R8], [R9]. Please revise this portion of text.
Author response: We appreciate the reviewer’s comments. We ignored former researches in drafting. Thanks for pointing out that.
Author action: Inappropriate expressions was revised in line 73-80, and [R8],[R9] were added as reference.
Concern #4: Every channel-related indicator (e.g., ToF, RSSI, CIR, CSI) depends on the environment. Please revise and discuss the portion of text at lines 79-83.
Author response: We appreciate the reviewer’s comments. For this topic, there are two reasons:1) The distribution of CIR changes with the environment~\cite{fernandez2023powergrid}, which means the features are unstable and not robust, therefore, not suitable for the KES scenario in which the vehicle transfers from place to place with uncertainty; 2) Compared to other channel-related indicators, e.g. ToF and RSS, CIR is more sensitive to the environment change because it contains all the information of the detailed multi-path in the environment. Different reflective surface composition or any surface moving in the region result in changes in CIR components, which happens very frequently on vehicle, e.g., different parking or pedestrian walking nearby.
Author action: We added the two reasons above into the article, line 95-102.
Concern #5: The problem discussed at line 92 is common to other neural-based methods. Please comment.
Author response: We appreciate the reviewer’s comments. We considered that compared to other networks that outputs a single coordinate, our model gives not only accurate coordinate but also reasonable corrected distances, which makes it easier to explain why our model is working in calculating the localization coordinates.
Author action: The texts were revised.
Concern #6: Lines 96-98 are too generic: please clarify and be more specific.
Author response: We appreciate the reviewer’s comments.
Author action: The paragraph is expanded and introduce the overall architecture of the model in line 129-139.
Concern #7: Before eq. (1): where is the origin of the coordinates? Is it relative to the vehicle? Not clear.
Author response: We appreciate the reviewer’s comments. The definition of the coordinate system was missing. Thanks for pointing out that.
Author action: Definition of the coordinate system was added before eq. (1).
Concern #8: Line 107-108: check the embedded equations for formal mathematical issues concerning scalar (in italics) and vectors (in bold). Check also superscript and subscript indexes.
Author response: We appreciate the reviewer’s comments. We have checked the equations and it is right.
Concern #9: Line 112: the MLE value is never defined.
Author response: We appreciate the reviewer’s comments. MLE definition is added in Eq.(3) and similarly, we added TDMAE definition in Eq.(18).
Concern #10: Lines 116-117: the method called Alternative Double-Sided Two Way Ranging is presented in the paper [R11] where eq. (3) is discussed in more details with respect to the application note [18]. Please use this new reference.
Author response: We appreciate the reviewer’s comments. The content of Decawave application note is poorer than [R11], so we followed the suggestion and replaced the reference.
Author action: The reference [18] is replaced with [R11].
Concern #11: Lines 123-131: is the ToF value computed by the nodes? Or by each node? Or by the system? How are the terms of eq. (3) acquired? Not clear, please rewrite.
Author response: We appreciate the reviewer’s comments. The detailed procedure is added in line 178-180.
Concern #12: Please check Sect. 3.3.1 for missing variable/vector definition and formal errors.
Author response: We appreciate the reviewer’s comments. The definition of r_sm is added and f_1, f_2 are set to be bold.
Concern #13: Sect. 3.3.2: the linear Least Squares Multi-lateration algorithm concerning the squared-range differences is well-known and the length of this section can be reduced. In addition, please insert here one or more references related to this algorithm, e.g. the Bensky textbook, and then simplify the equations (some equations have also formal errors as well).
Author response: We appreciate the reviewer’s comments. The derivation could be found easily in previous articles. However, we have to keep the calculation process since for method integrity and symbols are used even in shortened backpropagation feasibility comment.
Author action: Most of the derivation is deleted. Reference [21] and [29] were added.
Concern #14: Check eqs. (14) and (21) for formal errors.
Author response: We appreciate the reviewer’s comments. In Eq.(21), the vertical lines mean the absolute of the scalars and the subtraction is performed between scalars, so we think it is correct.
Author action: We fixed the formal errors in Eq.(14).
Concern #15: Sect. 3.4: given the results of [R11], since the LSM is a linear system, the cascade of a NN (that converges by hypothesis) and the LSM, converges as well. The proof is trivial given the results of [11]. Please insert the reference, provide a comment and then delete this section.
Author response: We appreciate the reviewer’s comments. [R12] is an excellent work of convergence proof. However, since the usage of the LSM in localization is not the same with a conversion before LSM itself, we have to easily confirm the conversion supports backpropagation.
Author action: Sect. 3.4 is deleted. In Sect. 3.3.2 line 204-208, simple comments on backpropagation is added. [R12] is added as reference.
Concern #16: Fig. 4 does not show any system. This figure summarizes the messages exchanged by the system devices. Please revise and add a figure about the proposed localization system.
Author response: We appreciate the reviewer’s comments. We didn’t mean to demonstrate the system appearance in fact since the vehicle is not published on market. We wanted to demonstrate how the system works and how data is collected.
Author action: The title of Fig.4 is replaced with “The flow chart of localization procedure in our KES system”.
Concern #17: Sect. 4.2 is unclear and confused. Please rewrite.
Author response: We appreciate the reviewer’s comments. Compared to other works, which usually used LiDAR and SLAM to acquire the groud-truth of the dataset, we only used manual tapes. We actually used marker pen to draw the contours which helps measure the coordinate of the testing points. So this Section introduces the dataset collecting process.
Author action: “convex hull “ was changed into “bounding rectangle”.
Concern #18: Lines 207-209: this part is unclear. Please revise.
Author response: We appreciate the reviewer’s comments. We trained the model in using (x,y,z) 3-d coordinates. However, because the KES only need to calculate the distance from keyfob to car’s bounding rectangle and height is ignored, so 2-d plane localization is important and enough for KES systems.
Author action: The detailed reason was added to line 249-253.
Concern #19: For a fair comparison of methods in Sect. 5, the parameters used by methods [8] and [13] must be included. Please add them to Tab. 3.
Author response: We appreciate the reviewer’s comments. Parameters used by comparative methods were added to Table.3. Relevant explanations were added in line 256-261.
Concern #20: Lines 223-237 and Fig.8: these parts are unclear and must be rewritten.
Author response: We appreciate the reviewer’s comments. This part demonstrates the effectiveness of DCNN. As shown in the figure, the cumulative distribution function (CDF) shows that after the DCNN, the distance error distributions is much smaller than the original ones. This helps explain why the model could give reasonable coordinates.
Author action: Full name of CDF was added.
Concern #21: Table 4: D and R are not defined.
Author response: We appreciate the reviewer’s comments.
Author action: D and R footnote were added.
Concern #22: Sect. 5.1: other localization methods, even high accuracy ones, are already available (see the literature and the end). For fair comparisons, results must be compared against state-of-the-art methods, e.g. [10], by using optimized parameters for each method.
Author response: We appreciate the reviewer’s comments. In our dataset, there is only distance measures and RSS measures, and no CIR signal is presented. Among the methods using these two features, we selected the most representative [8] as the numerical method and [13] as the deep learning method. Method using CIR including [R10] as you mentioned, could not be used on our dataset, as well as in vehicle scenario. This is mainly because the device needs to be low-energy and low-cost, and big networks can’t be deployed on this kind of device.
The performance of [R10] is RMSE=0.26. Comparatively, our method on our dataset is RMSE=0.18. Although there are two different datasets, the NLOS complexity of vehicle scenario is comparable or worse than that of [R10].
Author action: [R10] was not added as comparative method.
Concern #23: Sect. 5.3 and Tab. 4 are not clear and must be rewritten.
Author response: We appreciate the reviewer’s comments. We set this section to investigate the effects of D and R. Based on the results, we infer that D is more important in our method and R is effective in provide additional information missing in D. The results also support the conclusion that time-based localization mechanism is better than others in UWB systems.
Author action: Line 289-291 is rewritten for clear expression.
Concern #24: The bibliography is poor and many important references are missing. This section must be reconsidered and expanded. See also, but not limited to, the attached list.
Author response: We appreciate the reviewer’s comments. The reference you provided is very valuable and let us know more excellent works. This not only helped us improve the article, but also expanded our horizons.
Author action: Most of the reference from the attached list is added to the article.
Concern #25: Reference [5] is not written in English and no English translation is available. Please consider to delete it.
Author response: We appreciate the reviewer’s comments. Reference [5] is deleted.
Reviewer 2 Report
It is a very interesting combination of localization methods with neural networks. I recommend it for publication.
Author Response
We appreciate the reviewer for his kind remark.
Reviewer 3 Report
The paper is interesting and well-written. However, there are some suggestions for improvements.
It is also important to discuss why the given application requires low localization error to build further motivation for this work.
Please improve the use of superscripts. For example, on page 4, d_1^T is introduced, while the same variable T represents the transpose operation.
The proposed scheme's complexity in training data requirements and the manual effort required to collect the data should be quantified.
In section 3.3.2, the derivation ignores the estimation error. Why is this error term ignored, and what is the impact on the resulting algorithm?
It is also important to discuss why the given application requires low localization error to build further motivation for this work.
Almost all the figures are of poor quality. Please improve.
Author Response
We appreciate the editor and the reviewers for their valuable comments. The concerns raised by the reviewers have been addressed individually. The changes to the manuscript and our responses to the comments are detailed below. We sincerely hope that the revised manuscript has addressed the raised concerns and is acceptable for publication.
Response to Reviewer 3:
Two of your comments coincided by mistake and we only kept the latter one of them.
Concern #1: Please improve the use of superscripts. For example, on page 4, d_1^T is introduced, while the same variable T represents the transpose operation.
Author response: We appreciate the reviewer’s comments.
Author action: We replaced all the d^T with d^t.
Concern #2: The proposed scheme's complexity in training data requirements and the manual effort required to collect the data should be quantified.
Author response: We appreciate the reviewer’s comments. “It took three students one day to collect the dataset on a prototype vehicle. ” was added.
Concern #3: In section 3.3.2, the derivation ignores the estimation error. Why is this error term ignored, and what is the impact on the resulting algorithm?
Author response: We appreciate the reviewer’s comments. This whole derivation was deleted and references were added.
Concern #4: It is also important to discuss why the given application requires low localization error to build further motivation for this work.
Author response: We appreciate the reviewer’s comments. It is important to discuss why and thank you for pointing out that. For example, when the user is standing outside the driver's door attempting to open it, the KES need to accurately give a localization result that the keyfob is outside the vehicle, which requires the localization error to be lower than 30cm.
Author action: Text above about why the given application requires low localization error is added in line 22-25.
Concern #5: Almost all the figures are of poor quality. Please improve.
Author response: We appreciate the reviewer’s comments. We tried our best to provide high-quality figures, but most of the figures except Figure.7 and Figure.9 are of the best quality we could generate. Hope we can handle this problem in future editing and after acceptance.
Author action: Figure.7 and Figure.9 were replaced with high-resolution ones.
Round 2
Reviewer 1 Report
This Reviewer thanks the Authors for their time and effort spent in the revision of the manuscript. The new comments (in italics) are shown for each issue number as shown by the Authors. Finally, this Reviewer suggests the Authors to check thoroughly the manuscript for language errors and typos.
List of old issues:
Concern #1: There are a huge amount of papers about UWB localization using neural networks. Thus, the novelty of the proposed approach is questionable. High-resolution methods are also already available, e.g. [R1] at the end of these comments. For additional references of similar works using UWB and neural network processing-base methods see for instance [R2]-[R7], [10]. Thus, the manuscript contributions need to be reconsidered and rewritten.
Author response: We appreciate the reviewer’s comments. [R4] and[R5] used neural network to predict an indicator of the NLOS condition for further error mitigation. [R7] proposed two CNN networks using CIR to mitigate the ranging error and the localization error, respectively. [R2] estimate real distances. [R3], [R4], and [R6] estimated tag’s coordinate by NN. None of them have embedded numerical localization algorithm in the network architecture, therefore we think the embedded localization algorithm with neural network is a contribution of our work.
Author action: The recommended references are added in the revised paper and the contributions are revised.
➢ Strictly speaking, the Authors’ proposal is a DCNN that is trained according the loss function computed by the LSM method with the help of an additional layer of fixed coefficients that combines the corrected distances to provide the estimated one. A cascade of layers, with fixed or trainable coefficients, is a neural network. Therefore, the main contribution of the Authors is a kind of DCNN trained for localization purposes. It is obvious that the Authors’ proposal is a particular implementation of a deep neural network that must be paired with other implementations for analysis and comparison.
Concern #2: Topics about active localization are more complex than the brief discussion provided in Sect. 2. Please rewrite most parts of this section including also some comments regarding the works discussed in the seminal papers proposed in the Special Issue of the IEEE Signal Processing Magazine, July 2005 and the citing papers hereafter.
Author response: We appreciate the reviewer’s comments. The suggestion is valuable because the article is missing with overview of localization mechanism.
Author action: Those works were added in lines 55-60 for overview of localization mechanism and comment on different mechanisms in UWB systems.
➢ Ok.
Concern #3: Lines 66-79: works about UWB localization with neural networks have been available from at least 15 years: e.g. see [R8], [R9]. Please revise this portion of text.
Author response: We appreciate the reviewer’s comments. We ignored former researches in drafting. Thanks for pointing out that.
Author action: Inappropriate expressions was revised in line 73-80, and [R8], [R9] were added as reference.
➢ Ok.
Concern #4: Every channel-related indicator (e.g., ToF, RSSI, CIR, CSI) depends on the environment. Please revise and discuss the portion of text at lines 79-83.
Author response: We appreciate the reviewer’s comments. For this topic, there are two reasons:1) The distribution of CIR changes with the environment~\cite{fernandez2023powergrid}, which means the features are unstable and not robust, therefore, not suitable for the KES scenario in which the vehicle transfers from place to place with uncertainty; 2) Compared to other channel-related indicators, e.g. ToF and RSS, CIR is more sensitive to the environment change because it contains all the information of the detailed multi-path in the environment. Different reflective surface composition or any surface moving in the region result in changes in CIR components, which happens very frequently on vehicle, e.g., different parking or pedestrian walking nearby.
Author action: We added the two reasons above into the article, line 95-102.
➢ The influence of the environment on the measurements (and the features, accordingly) is complex and thousands of papers have been written about this topic. This Reviewer understands the point of view of the Authors even if it is worth mentioning that RSS can be obtained by exploiting mean CSI measurements, too. This problem deserves additional attentions that cannot be summarized here. In any case, for future works, this Reviewer suggests the Authors to investigate also the processing of CSI-based features.
Concern #5: The problem discussed at line 92 is common to other neural-based methods. Please comment.
Author response: We appreciate the reviewer’s comments. We considered that compared to other networks that outputs a single coordinate, our model gives not only accurate coordinate but also reasonable corrected distances, which makes it easier to explain why our model is working in calculating the localization coordinates.
Author action: The texts were revised.
➢ Ok.
Concern #6: Lines 96-98 are too generic: please clarify and be more specific.
Author response: We appreciate the reviewer’s comments.
Author action: The paragraph is expanded and introduce the overall architecture of the model in line 129-139.
➢ Ok.
Concern #7: Before eq. (1): where is the origin of the coordinates? Is it relative to the vehicle? Not clear.
Author response: We appreciate the reviewer’s comments. The definition of the coordinate system was missing. Thanks for pointing out that.
Author action: Definition of the coordinate system was added before eq. (1).
➢ Ok.
Concern #8: Line 107-108: check the embedded equations for formal mathematical issues concerning scalar (in italics) and vectors (in bold). Check also superscript and subscript indexes.
Author response: We appreciate the reviewer’s comments. We have checked the equations and it is right.
➢ Ok. Regarding the mathematical aspects, this Reviewer encourages also the Authors to exploit a consistent and common way to define matrices/vectors using for all vectors/matrices the squared parentheses such as x = [x1, x2, …, xN]T for the generic column vector x of size N x 1. It is also important to define the size of each vector/matrix in terms of number of rows and number of columns.
Concern #9: Line 112: the MLE value is never defined.
Author response: We appreciate the reviewer’s comments. MLE definition is added in Eq.(3) and similarly, we added TDMAE definition in Eq.(18).
➢ Ok.
Concern #10: Lines 116-117: the method called Alternative Double-Sided Two Way Ranging is presented in the paper [R11] where eq. (3) is discussed in more details with respect to the application note [18]. Please use this new reference.
Author response: We appreciate the reviewer’s comments. The content of Decawave application note is poorer than [R11], so we followed the suggestion and replaced the reference.
Author action: The reference [18] is replaced with [R11].
➢ Ok.
Concern #11: Lines 123-131: is the ToF value computed by the nodes? Or by each node? Or by the system? How are the terms of eq. (3) acquired? Not clear, please rewrite.
Author response: We appreciate the reviewer’s comments. The detailed procedure is added in line 178-180.
➢ Ok. Please also remember to use ToF instead of tof through all the manuscript.
Concern #12: Please check Sect. 3.3.1 for missing variable/vector definition and formal errors.
Author response: We appreciate the reviewer’s comments. The definition of r_sm is added and f_1, f_2 are set to be bold.
➢ Ok.
Concern #13: Sect. 3.3.2: the linear Least Squares Multi-lateration algorithm concerning the squared-range differences is well-known and the length of this section can be reduced. In addition, please insert here one or more references related to this algorithm, e.g. the Bensky textbook, and then simplify the equations (some equations have also formal errors as well).
Author response: We appreciate the reviewer’s comments. The derivation could be found easily in previous articles. However, we have to keep the calculation process since for method integrity and symbols are used even in shortened backpropagation feasibility comment.
Author action: Most of the derivation is deleted. Reference [21] and [29] were added.
➢ Ok.
Concern #14: Check eqs. (14) and (21) for formal errors.
Author response: We appreciate the reviewer’s comments. In Eq.(21), the vertical lines mean the absolute of the scalars and the subtraction is performed between scalars, so we think it is correct.
Author action: We fixed the formal errors in Eq.(14).
➢ This concern was related to the use of |.| instead of the norm ||.||. It is fine to use LE(.) that refers to the norm ||.||.
Concern #15: Sect. 3.4: given the results of [R11], since the LSM is a linear system, the cascade of a NN (that converges by hypothesis) and the LSM, converges as well. The proof is trivial given the results of [11]. Please insert the reference, provide a comment and then delete this section.
Author response: We appreciate the reviewer’s comments. [R12] is an excellent work of convergence proof. However, since the usage of the LSM in localization is not the same with a conversion before LSM itself, we
have to easily confirm the conversion supports backpropagation.
Author action: Sect. 3.4 is deleted. In Sect. 3.3.2 line 204-208, simple comments on backpropagation is added. [R12] is added as reference.
➢ It is fine to include some comments related to backpropagation, even if it is also worth mentioning here that the LSM method is a quadratic optimization problem with well-known convergence properties.
Concern #16: Fig. 4 does not show any system. This figure summarizes the messages exchanged by the system devices. Please revise and add a figure about the proposed localization system.
Author response: We appreciate the reviewer’s comments. We didn’t mean to demonstrate the system appearance in fact since the vehicle is not published on market. We wanted to demonstrate how the system works and how data is collected.
Author action: The title of Fig.4 is replaced with “The flow chart of localization procedure in our KES system”.
➢ Ok.
Concern #17: Sect. 4.2 is unclear and confused. Please rewrite.
Author response: We appreciate the reviewer’s comments. Compared to other works, which usually used LiDAR and SLAM to acquire the groud-truth of the dataset, we only used manual tapes. We actually used marker pen to draw the contours which helps measure the coordinate of the testing points. So this Section introduces the dataset collecting process.
Author action: “convex hull “ was changed into “bounding rectangle”.
➢ Ok.
Concern #18: Lines 207-209: this part is unclear. Please revise.
Author response: We appreciate the reviewer’s comments. We trained the model in using (x,y,z) 3-d coordinates. However, because the KES only need to calculate the distance from keyfob to car’s bounding rectangle and height is ignored, so 2-d plane localization is important and enough for KES systems.
Author action: The detailed reason was added to line 249-253.
➢ Ok.
Concern #19: For a fair comparison of methods in Sect. 5, the parameters used by methods [8] and [13] must be included. Please add them to Tab. 3.
Author response: We appreciate the reviewer’s comments. Parameters used by comparative methods were added to Table.3. Relevant explanations were added in line 256-261.
➢ Ok.
Concern #20: Lines 223-237 and Fig.8: these parts are unclear and must be rewritten.
Author response: We appreciate the reviewer’s comments. This part demonstrates the effectiveness of DCNN. As shown in the figure, the cumulative distribution function (CDF) shows that after the DCNN, the distance error distributions is much smaller than the original ones. This helps explain why the model could give reasonable coordinates.
Author action: Full name of CDF was added.
➢ The concern was not related to the CDF but related to the meaning of the “original” and “corrected” terms in the text and in the figure. The Authors must specify what the original and corrected methods are.
Concern #21: Table 4: D and R are not defined.
Author response: We appreciate the reviewer’s comments.
Author action: D and R footnote were added.
➢ Ok. However, the note numbers introduce some confusions due to numbers that are similar to exponents. This Reviewer suggests the Authors to define D and R in the text or in the table’s legend.
Concern #22: Sect. 5.1: other localization methods, even high accuracy ones, are already available (see the literature and the end). For fair comparisons, results must be compared against state-of-the-art methods, e.g. [10], by using optimized parameters for each method.
Author response: We appreciate the reviewer’s comments. In our dataset, there is only distance measures and RSS measures, and no CIR signal is presented. Among the methods using these two features, we selected the most representative [8] as the numerical method and [13] as the deep learning method. Method using CIR including [R10] as you mentioned, could not be used on our dataset, as well as in vehicle scenario. This is mainly because the device needs to be low-energy and low-cost, and big networks can’t be deployed on this kind of device. The performance of [R10] is RMSE=0.26. Comparatively, our method on our dataset is RMSE=0.18. Although there are two different datasets, the NLOS complexity of vehicle scenario is comparable or worse than that of [R10].
Author action: [R10] was not added as comparative method.
➢ Ok. In any case, this Reviewer suggests the Authors to include the RMSE value (i.e, 0.26 [m]) as a main result of reference [10] (and other methods if possible) in the text as a comparison w.r.t. the proposed method (with caveats and brief comments).
Concern #23: Sect. 5.3 and Tab. 4 are not clear and must be rewritten.
Author response: We appreciate the reviewer’s comments. We set this section to investigate the effects of D and R. Based on the results, we infer that D is more important in our method and R is effective in provide additional information missing in D. The results also support the conclusion that time-based localization mechanism is better than others in UWB systems.
Author action: Line 289-291 is rewritten for clear expression.
➢ Ok. This Reviewer suggests also to check the language of Sect. 5.4 for unclear phrases such as those at lines 304-307 “An extreme circumstance when α is set to be 1.0, in which case the model is actually set to fit the ranging error only, results in an MLE error bigger than that with α set to 0.9, which disapproves the opinion of the studies minimizing distance errors [15,26]” and lines 316-318 “the model could perform better than the comparative methods on the MLE metric. This proves that the proposed model is robust in α hyperparameter setting”
Concern #24: The bibliography is poor and many important references are missing. This section must be reconsidered and expanded. See also, but not limited to, the attached list.
Author response: We appreciate the reviewer’s comments. The reference you provided is very valuable and let us know more excellent works. This not only helped us improve the article, but also expanded our horizons.
Author action: Most of the reference from the attached list is added to the article.
➢ Ok.
Concern #25: Reference [5] is not written in English and no English translation is available. Please consider to delete it.
Author response: We appreciate the reviewer’s comments. Reference [5] is deleted.
➢ Ok.
Author Response
We appreciate the editor and the reviewers for their valuable comments. The concerns raised by the reviewers have been addressed individually. The changes to the manuscript and our responses to the comments are detailed below, omitting concerns already addressed. Major revisions are marked in blue in the pdf file, too. We sincerely hope that the revised manuscript has addressed the raised concerns and is acceptable for publication.
Q1: Strictly speaking, the Authors’ proposal is a DCNN that is trained according the loss function computed by the LSM method with the help of an additional layer of fixed coefficients that combines the corrected distances to provide the estimated one. A cascade of layers, with fixed or trainable coefficients, is a neural network. Therefore, the main contribution of the Authors is a kind of DCNN trained for localization purposes. It is obvious that the Authors’ proposal is a particular implementation of a deep neural network that must be paired with other implementations for analysis and comparison.
Author response: We appreciate the reviewer’s suggestions. We indeed didn’t analyze the difference between our method and the related works in the former manuscript. We added a clearer comparison between the related works and our method in Sect. 2, lines 127-136. Hopefully, the contribution of our work can be clearer with this comparison.
Q4: The influence of the environment on the measurements (and the features, accordingly) is complex and thousands of papers have been written about this topic. This Reviewer understands the point of view of the Authors even if it is worth mentioning that RSS can be obtained by exploiting mean CSI measurements, too. This problem deserves additional attentions that cannot be summarized here. In any case, for future works, this Reviewer suggests the Authors to investigate also the processing of CSI-based features.
Author response: We appreciate the reviewer’s suggestions. In the UWB signal, CSI often refers to the CIR. We also consider CIR a good feature to represent the NLOS condition. Only in the present research, we didn’t find a good way to exploit it in the KES system. However, in future works, we are going to explore more effective ways to exploit the CIR feature. We added this as future work in Sect.6.
Q8: Regarding the mathematical aspects, this Reviewer encourages also the Authors to exploit a consistent and common way to define matrices/vectors using for all vectors/matrices the squared parentheses such as x = [x1, x2, …, xN]T for the generic column vector x of size N x 1. It is also important to define the size of each vector/matrix in terms of number of rows and number of columns.
Author response: We appreciate the reviewer’s suggestions. We revised some of the definitions of vectors to be surrounded by squared parentheses.
Q11: Ok. Please also remember to use ToF instead of tof through all the manuscript.
Author response: We appreciate the reviewer’s comments. Based on your comment, we checked and revised the ToF abbreviation throughout the article.
Q14: This concern was reluse of |.| instead of the norm ||.||. It is fine to use LE(.) that refers to the norm ||.||.
Author response: We appreciate the reviewer’s comments. LE(.) is actually defined as the Euclidean distance between the ground-truth coordinate and the estimated one, so we use the norm to represent the Euclidean distance. In TDAE(.), however, the |.| refers to the absolute error of distances(scalars), instead of the norm between the vectors.
Q15: It is fine to include some comments related to backpropagation, even if it is also worth mentioning here that the LSM method is a quadratic optimization problem with well-known convergence properties.
Author response: We appreciate the reviewer’s comments. We added the brief comment on LSM here, labeled by blue in lines 213-214.
Q20: The concern was not related to the CDF but related to the meaning of the “original” and “corrected” terms in the text and in the figure. The Authors must specify what the original and corrected methods are.
Author response: We appreciate the reviewer’s comments. The ‘Original’ actually refers to the raw distance data in the dataset without correction. We added the explanation in the Table’s legend.
Q21: Ok. However, the note numbers introduce some confusions due to numbers that are similar to exponents. This Reviewer suggests the Authors to define D and R in the text or in the table’s legend.
Author response: We appreciate the reviewer’s comments. We revised the manuscript and define the D and R in the table’s legend.
Q22: Ok. In any case, this Reviewer suggests the Authors to include the RMSE value (i.e, 0.26 [m]) as a main result of reference [10] (and other methods if possible) in the text as a comparison w.r.t. the proposed method (with caveats and brief comments).
Author response: We appreciate the reviewer’s comments. We added the result of reference [10] in the text, lines 297-304, as well as the RMSE and TDMAE testing results of our method.
Q23: Ok. This Reviewer suggests also to check the language of Sect. 5.4 for unclear phrases such as those at lines 304-307 “An extreme circumstance when α is set to be 1.0, in which case the model is actually set to fit the ranging error only, results in an MLE error bigger than that with α set to 0.9, which disapproves the opinion of the studies minimizing distance errors [15,26]” and lines 316-318 “the model could perform better than the comparative methods on the MLE metric. This proves that the proposed model is robust in α hyperparameter setting”
Author response: We appreciate the reviewer’s comments. We rewrote this part of the text in Sect. 5.4 for clearer expressions.